behaviour/cognition

acceptance threshold, egg discrimination, perceptual series, recognition systems

**Author for correspondence:**
Mark E. Hauber
e-mail: mhauber@illinois.edu

# The limits of egg recognition: testing acceptance thresholds of American robins in response to decreasingly egg-shaped objects in the nest

Mark E. Hauber[1,2,3], Sarah K. Winnicki[1,2], Jeffrey P. Hoover[3], Daniel Hanley[4] and Ian R. Hays[5]

[1]Department of Evolution, Ecology, and Behavior, [2]Program in Ecology, Evolution, and Conservation, and [3]Illinois Natural History Survey, Prairie Research Institute, University of Illinois at Urbana-Champaign, Urbana, IL, USA
[4]Department of Biology, George Mason University, Fairfax, VA, USA
[5]Department of Biological Sciences; Rutgers, The State University of New Jersey, Newark, NJ, USA

MEH, 0000-0003-2014-4928; DH, 0000-0003-0523-4335

Some hosts of avian brood parasites reduce or eliminate the costs of parasitism by removing foreign eggs from the nest (rejecter hosts). In turn, even acceptor hosts typically remove most non-egg-shaped objects from the nest, including broken shells, fallen leaves and other detritus. In search for the evolutionary origins and sensory mechanisms of egg rejection, we assessed where the potential threshold between egg recognition and nest hygiene may lie when it comes to stimulus shape. Most previous studies applied comparisons of egg-sized objects with non-continuous variation in shape. Here, instead, we used two series of three-dimensional-printed objects, designed *a priori* to increasingly diverge from natural eggs along two axes (width or angularity) of shape variation. As predicted, we detected transitions from mostly acceptance to mostly rejection in the nests of American robins *Turdus migratorius* along each of the two axes. Our methods parallel previous innovations in egg-rejection studies through the use of continuous variation in egg coloration and maculation contrast, to better understand the sensory limits and thresholds of variation in egg recognition and rejection in diverse hosts of avian brood parasites.

# 1. Introduction

Most birds keep their nests relatively clean, at least during the incubation stage, by removing detritus (e.g. fallen leaves or flower petals) and broken eggs/shells from the nest cup [1]; such nest hygiene behaviour has been the focus of extensive research, including some of the classic work by Nobel-prize winning Niko Tinbergen [2,3]. Through removing foreign objects from the nest, parent birds can gain fitness benefits by increasing incubation efficiency and reducing infection or other dangers to incubation success (e.g. [4,5]).

Hosts of avian brood parasites exhibit a wide range of egg-rejection abilities [6,7], which provides a tractable system for studying how birds recognize eggs from non-egg objects. By removing the parasitic egg(s), hosts forgo reproductive costs accrued when raising unrelated chick(s) in their brood [8,9]. However, egg recognition can be complex as there are both costly trade-offs and physical-cognitive constraints accompanying this behaviour. First, egg rejection can be costly because hosts may mistakenly reject (rejection error: [10,11]) or damage their own eggs (rejection cost: [12,13]) when attempting to pierce and remove thicker-shelled parasitic eggs [14,15]. Second, hosts are constrained by their own sensory inabilities and may be unable to recognize mimetically coloured and patterned parasite eggs [16,17]. Finally, hosts may face mechanical constraints if they are physically unable to pierce or grasp and lift foreign eggs too thick or heavy from the nest [18,19]. How hosts of brood parasites balance the benefits and costs of removing foreign versus own eggs from the nest remains an area of active research [20].

Central to this line of research is a more general question: how and when do parent birds recognize a foreign egg-shaped object as a direct indicator of brood parasitism versus as a cue of debris in the nest [5,21]? Previous studies on the evolutionary origins, developmental precursors and/or sensory triggers of egg-rejection behaviours have repeatedly confirmed that foreign-egg rejection and detritus-removal are cognitively different and independent behaviours; typically, these works found a lack of covariation between the rejection rates of detritus versus foreign eggs from the nest (e.g. in a non-egg-rejecter host: [22]; in a rejecter host: [23], but also see [24]).

Recent methodological advances owing to custom-designed three-dimensional printing of model eggs and other stimuli, along with a predetermined fine-scale variation in the shape of egg-like objects, provide novel avenues to explore classic questions. Initial research by Igic *et al.* [25] explored whether subtle variation in egg shape (within 10% of the natural shape metrics) predicted egg-rejection behaviours. More recently, Yang *et al.* [26] expanded on this research by including two different series of eggs that had graded variation from natural egg-shaped to unnatural, non-egg-like objects, as stimuli for egg rejection. The study by Yang *et al.* [26] also included a set of exposures of egg-like and two, incrementally non-egg-like models to a predominantly egg-acceptor host of the common cuckoo *Cuculus canorus*, the barn swallow *Hirundo rustica* in China, demonstrating experimentally that greater deviation from natural egg shape increased rejection rates.

Here we used these technological and design advances to study the rejection responses of American robins *Turdus migratorius*, a robust egg-rejecter host species of obligate brood parasitic brown-headed cowbirds *Molothrus ater* in North America (figure 1a). Specifically, we used both of the different directions of egg-shape manipulations developed by Yang *et al.* [26] to generate two series (consisting of 5–6 steps; figure 1b) of increasingly non-egg-like models to characterize the response curve's slope to assess the position of various egg-rejection thresholds (*sensu* [27,28]) along with quasi-continuous gradients of model egg traits used as egg-rejection cues (e.g. [29,30]).

# 2. Methods

We sourced models from two different origins: for natural, egg-shaped objects, we used three-dimensional-printed, brown-headed cowbird-sized and -weighted model eggs (sourced from www. Shapeways.com: catalogue name 'cow bird', in versatile natural plastic; for production and dimension details, see [25]). For the continuous series of decreasingly egg-shaped models, we generated the two different sets of designs:

(1) we manufactured six different models similar to Yang *et al.* [26] 'surface edge' variable (their fig. 1, top rows), which increased the length of the flat panels that are used to generate a decreasingly egg-like shape (this is our 'panel-length' treatment, figure 1b, top row); and

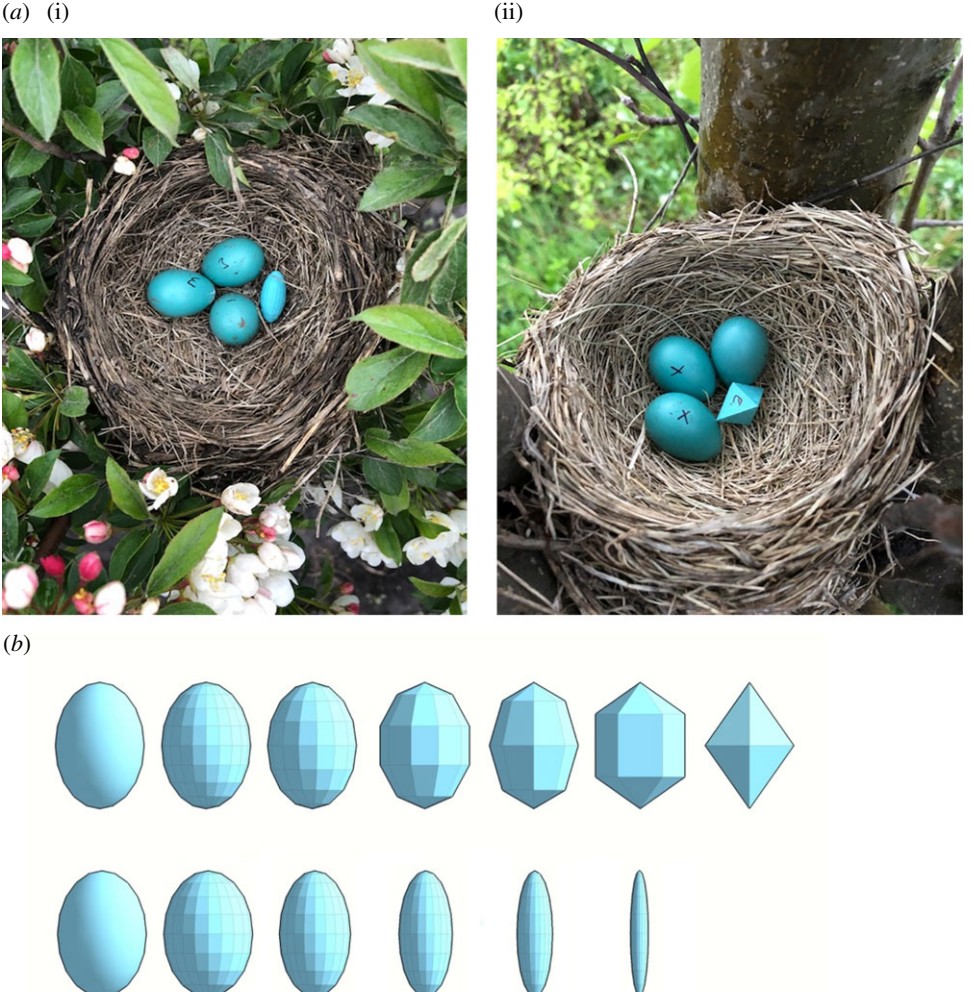

**Figure 1.** (*a*) Representative experimental clutches of American robins with a width (i) or panel-length (ii) model in each nest, painted to match the colour of the host's own eggs (photo credits: M. Hauber); (*b*) the designs of the two series (upper row is panel-length (mm): 0, 3.7, 4.8, 7.4, 8.8, 11.7 and 12.6; lower row is width (mm): 17.1, 17.1, 13.9, 10.3, 6.9 and 3.1) of model shapes used for this experiment (the leftmost eggs are the controls (Shapeway models, from 2019), the rest are the Voxel-Magic models (from 2020).

(2)   we generated five different models similar to Yang *et al*. [26] 'stereoscopic structure' variable (their fig. 1, bottom rows), which gradually decreased the width but not the length of the egg-shaped structure (this is our 'width' treatment, figure 1*b*, bottom row).

Both (1) and (2) were produced using the custom-made services of www.Voxel-Magic.com, again in versatile natural plastic. The length (22.8 mm) and width (17.1 mm) for the panel-length treatment model was set to match that of our control model egg; similarly, the length (22.8 mm) of the models in the width treatment was set to the control egg. We characterized each model by measuring the length of its linear edge segment for the panel-length treatment (it was set to 0 for the smooth control egg), and by measuring the width of the models at their broadest point for the width treatment (for details, figure 1*b*). Measurements were made using a caliper to the nearest 0.1 mm. All models were painted a robin-like blue colour (paint-mix details sourced from [31]), in triplicate coats which also generated a standardized surface consistency. The resulting avian-perceived chromatic dissimilarity (just noticeable difference: JND) from natural robin egg colours was approximately 2 JND for the mimetic eggs [32,33].

We sourced published data from Hauber *et al*. [32] for the egg-rejection rates of robin-blue mimetically painted control eggs (one rejection out of *n* = 15 trials). To deploy the two new series of model eggs, during May–June, 2020, we located active nests of free-living American robins, with permission, throughout private tree farms, and gardens, near Urbana, IL, USA (for details of the study area and search methods, see [34]. A nest was considered active if (i) the clutch size grew on consecutive days,

(ii) a robin was flushed from the nest, and/or (iii) robin eggs were warm to human touch. These methods yielded nests for experimental treatments during the laying and incubation stages, for which the data were combined based on a lack of difference in egg-rejection rates between these stages during our own prior experiments on robins at this study area [35].

Adult robins were not captured or marked in this study, and we used each nest as the unit of biological and statistical analysis. We conducted these experiments in as many simultaneously active nests as possible at four separate study sites (more than 5 km from each other), thus reducing the potential for pseudoreplication owing to studying the same breeding robin(s) at repeated breeding attempts throughout the season.

Once an active robin nest was located, it was exposed to 1–4 (median: 2) treatments/nest, each trial with a randomly chosen model selected from across both model series ($n = 5$–9/model type). We revisited each nest 1 day after the model deployment and assessed whether the model egg was present (accepted) or missing (rejected) (as per [33]). We then initiated the next trial with a randomly chosen treatment at the same nest ($n = 79$ total trials in 2020). Depredated (all eggs missing or broken; 8%) or abandoned (nests with cold eggs on two consecutive days; 10%) nests were removed from the datasets prior to analyses, as nest abandonment is not a response by American robins to experimental brood parasitism ([36]; for each treatment's sample size, see figure 2 legend). We detected no recognition errors as none of the robin's own eggs went missing from active nests (*sensu* [32,37]).

We separated the data for each of the two series in our statistical analyses, combining each with data from control eggs, and used two-tailed general linear mixed models with binomial distribution in the 'lme4' package [38] of the R Statistical Program [39]. The response variable was the bivariate outcome of each trial (accepted/rejected), with model-series trait metric (panel-length or width) (each as a continuous trait) and trial number (whether 1st, 2nd, etc. per each nest) of the experiment (also as a continuous trait) included as predictor variables and nest ID used as a random effect. We also repeated these same analyses for the 2020 datasets alone, excluding control eggs from 2019. Note that data from one of the 2020 egg types (panel-length: 3.65 mm, width: 17.1 mm, $N = 8$ trials) were used for both the panel-length and width analyses. We set $\alpha = 0.05$.

These studies were approved by the animal ethics committee (IACUC) of the University of Illinois at Urbana-Champaign (#17259), and by USA federal (MB08861A-3) and Illinois state agencies (NH19.6279). The data used for this article can be accessed at Figshare.com and its doi:10.6084/m9.figshare.13469832.

## 3. Results

Regarding the panel-length model series treatment, we found a shift from mostly acceptance to mostly rejection as panel-length increased, but did so significantly only when the control eggs were included ($z = 2.35$, $p = 0.019$, odds ratio: 0.75; figure 2a) and not when these were excluded ($z = 1.60$, $p = 0.11$; odds ratio: 0.76). There were non-significant effects of increased rejection with greater trial number at each nest in both datasets ($z = 1.85$, $p = 0.065$, odds ratio: 0.59, and $z = 1.58$, $p = 0.11$, odds ratio: 0.59, respectively).

Regarding the width model series, we found a shift from mostly acceptance to mostly rejection as width decreased and did so significantly both when the control eggs were included ($z = -3.52$, $p = 0.00043$, odds ratio: 1.62; figure 2b) or excluded ($z = -3.04$, $p = 0.0024$, odds ratio: 1.65) (figure 2b). There were significant effects of more rejections with increasing trial numbers at each nest in both datasets ($z = 2.89$, $p = 0.0038$, odds ratio: 0.47 and $z = 2.50$, $p = 0.012$, odds ratio: 0.45, respectively).

## 4. Discussion

Using a continuous series of egg-shape manipulations with three-dimensional-printed models (*sensu* [26]), we were able to identify two different series of responses by American robins, each with its specific response curvatures (steepness/shallowness and the location of the inflection point) of the logistic regression function, in response to our two treatment series (figure 2). There appeared to be a steep and significant drop in egg-acceptance rates as models become narrower at the equator (width series), in both the dataset that included the control model eggs and the one that did not, whereas there was a shallower response function when altering the linearity of the curvature of the egg models (panel-length series), and this was only significant when the expanded (control-included) dataset was analysed (figure 2). These relative patterns and slope comparisons remained consistent when plotting

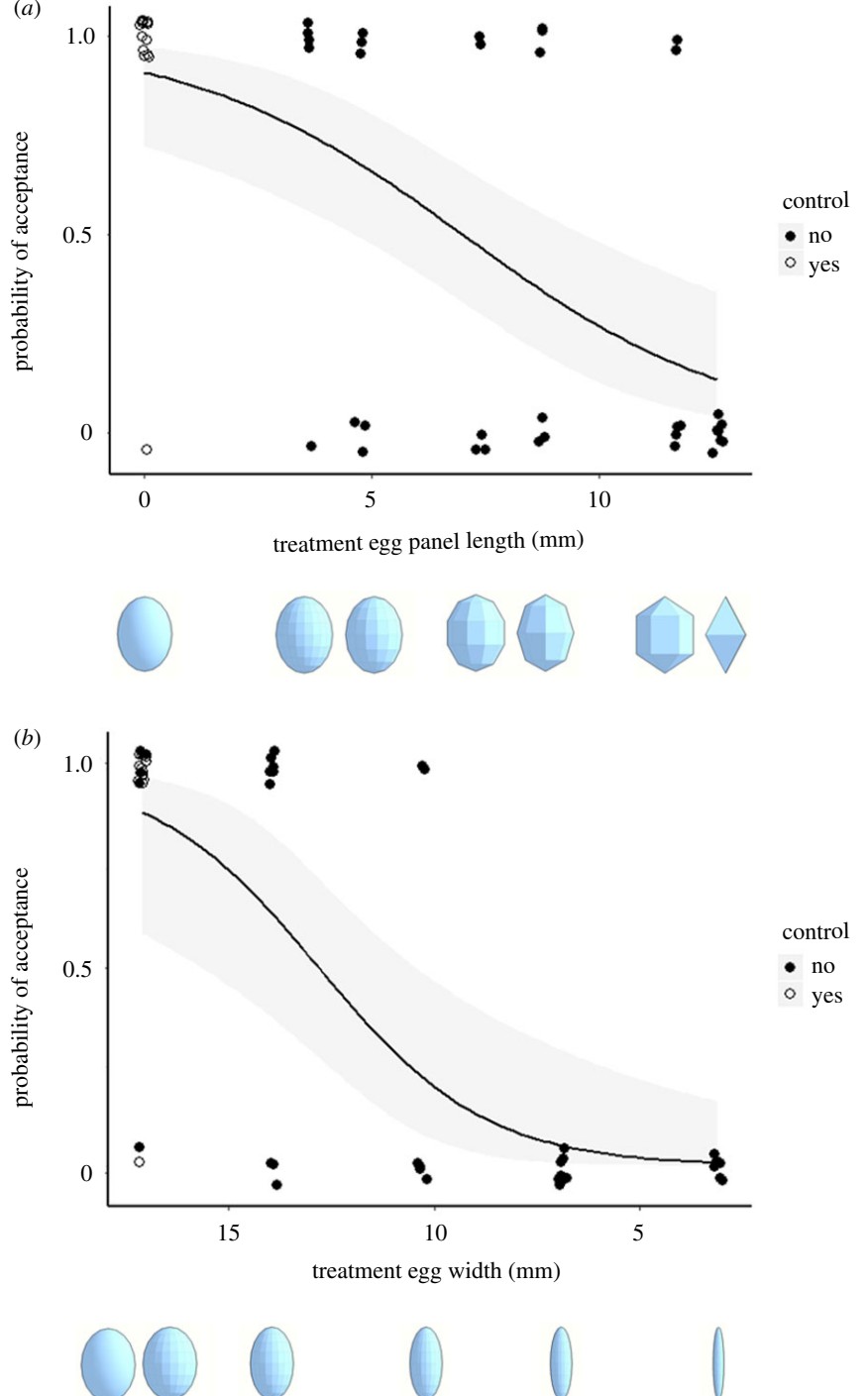

**Figure 2.** Egg-rejection responses to the two different, continuous series of model egg-shape versions: (a) panel-length series (sample sizes from left to right: $n = 15$ (control), 5, 6, 5, 6, 6 and 7); (b) width series ($n = 15$ (control), 5, 9, 6, 9 and 6). The white circles represent the control model cowbird eggs (from 2019) in each plot. Shaded areas represent the 95% confidence intervals around the logistic regression curves (black lines).

the logistic regression curves against standardized ($z$-score transformed) data on the x-axes between the two model series (electronic supplementary material, figure S1).

Critically, there were several $p$-values that hovered around our $\alpha = 0.05$ value; nonetheless, the odds ratio calculations confirmed that the relative biological impact of the two experimentally manipulated egg-trait metrics were consistent, in that less egg-shaped models were rejected more often. In turn, in both series of trials, we also found that the patterns of increased rejection rates with higher trial numbers were variably significant, but presented consistently similar odds ratios, implying that biologically and

statistically, future analyses of repeated experimentation at the same robin nests should consistently account for individual's prior experience with natural or experimental parasitism (*sensu* [40–42]).

On the one hand, our results are surprising, as egg-widths (and, hence, aspect-ratios) are naturally variable eggshell traits in robins [33] and other calcareous egg-laying species [43], whereas linear edges with sharp angles are altogether absent in natural avian egg shapes [44]; therefore, we would have expected a shallower and later rise in egg-rejection rates along the former (width) series of shapes. On the other hand, it may be that robins can discriminate better the variation in width (and aspect) than in panel-lengths because the former ones *are* in fact the traits that naturally vary between robin and cowbird eggs [25,33], and the perception of these two egg types could be under selective pressure in the context of robin's egg recognition processes. In turn, robins never face eggs with sharp angles in nature, and so it may be that this variable's perception is not under selection pressure to be recognized and/or discriminated and, thus, is more difficult for robins to perceive and assess. Finally, we detected no parallels between the increased rejection of decreasingly egg-shaped objects and higher rates of the removal of the robin's own eggs, as we recorded no recognition errors through the disappearance of the hosts' own eggs (*sensu* [32,37]).

We cannot quantitatively directly compare our results with those of Yang *et al*. [26], because they

(i) studied a different host species (theirs: barn swallow versus ours: American robin) and their natural brood parasites (common cuckoos versus brown-headed cowbirds),
(ii) sampled fewer steps (*n* = 3 each versus *n* = 5 and 6 each) along the two axes of egg-shape variability (their fig. 4 versus our figure 1*b*), and
(iii) used a non-mimetic egg colour (green) relative to the host or the parasitic egg (versus we used mimetic robin-blue painted model eggs).

Qualitatively, however, we can still report two similarities between their study and ours. First, we both detected greater rejection rates between control and increasingly non-egg-shaped models. Second, the rejection rates of both their and our width series rose more sharply than the panel-length series, with greater deviation from the control. Future work in the swallow system could deploy again both series of model eggs, but this time painted to be mimetic of the host eggs. However, non-mimetically (cowbird-like) painted model egg series in the American robin system are not likely to produce much variation with respect to rejection, since cowbird-like eggs and painted egg-shaped models are already rejected at over 90% of robin nests [32].

We need to acknowledge explicitly that in our model designs, there are two potential confounds that are known from the prior egg-rejection studies that may also explain our findings of greater egg rejection: smaller volume (e.g. [45]) and decreasing mass (e.g. [46]). Specifically, the less egg-like the shapes in both of our model series, the smaller their volumes and their weights, which may make it even easier for robins to reject these models. However, prior work using colour manipulations of model eggs demonstrated that robins are able to reject egg models that are the same size, weight and shape as cowbird eggs at greater than or equal to 90% when they are coloured and patterned similarly to natural cowbird eggs (e.g. [32,47]), implying that further decreases in volume and/or mass are not necessary for robins to reach near 100% rejection relative to cowbird-sized models. Future work should, nonetheless, standardize the weight of the less egg-shaped models in both series, perhaps by inserting pieces of lead into the models. However, the model volume cannot be altered without also altering the length of models.

Recent published works on using continuous natural eggshell variation (e.g. colour [29], spot-contrast [48], pixel resolution [30]) have improved upon earlier artificial [49] or categorical [50] variation in model egg appearances. The more recent studies also provide increasingly detailed and quantitative information about not only where egg-acceptance thresholds lie (e.g. [51,52]), but also whether (e.g. [40]) and how much they shift, and in which direction, with changing stimulus salience and multiple recognition cue components [53,54]. Here, the use of more extended shape series (relative to [26]) allowed us to estimate the curvature of the egg-rejection logistic function along with our gradients of shape variation, making these new results comparable with the recent series of egg-rejection experiments using continuous model (egg) trait variation(s) as predictor(s) of egg-rejection response patterns (*sensu* [31]).

Ethics. These studies were approved by the animal ethics committee (IACUC) of the University of Illinois at Urbana-Champaign (no. 17259), and by USA federal (MB08861A-3) and Illinois state agencies (NH19.6279).

Data accessibility. The data used for this article can be accessed at Figshare.com and its doi:10.6084/m9.figshare.13469832.

Authors' contributions. M.E.H. conceived the study, conducted partial fieldwork, analysed the data and wrote the first draft. S.K.W. conducted partial fieldwork, visualized the data and edited the manuscript. J.P.H. conducted partial

fieldwork and edited the manuscript; D.H. assisted with the methodology and edited the manuscript; I.R.H. designed the research stimuli, visualized the data and edited the manuscript.

Competing interests. We declare we have no competing interests.

Funding. Funding for this project was provided by the Harley Jones Van Cleave Professorship (to M.E.H.) and an Illinois Distinguished Fellowship (to S.K.W.). Additional funding during the preparation of this article was supplied by the Hanse-Wissenschaftskolleg (Institute for Advanced Study) in Delmenhorst, Germany (to M.E.H.).

Acknowledgements. We thank the landowners around Urbana for granting access to their properties. Research permits were provided by institutional, state and federal agencies. For helpful assistance and careful comments, we thank A. Luro, V. Fiorini, M. Louder, H. Pollock, O. Rhodes. and the editors and referees of this journal.

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
