## [Reviewer comments · Royal Society Open Science]

Review History

RSOS-201615.R0 (Original submission)

Review form: Reviewer 1

Is the manuscript scientifically sound in its present form?

Yes

Are the interpretations and conclusions justified by the results?

No

Is the language acceptable?

Yes

Do you have any ethical concerns with this paper?

No

Have you any concerns about statistical analyses in this paper?

Yes

Recommendation?

Major revision is needed (please make suggestions in comments)

Comments to the Author(s)

This very interesting study tackles fundamental questions about discerning between nest hygiene and foreign egg rejection behavior in birds. The authors partly replicated methodologies of Igic et al. (2015) and Yang et al. (2019) by manipulating shape of 3D printed models on continuous scale from egg-shaped to non-egg like objects. Using the original analysis approach, they were able to detect transitions in behavioral responses according to decreasing egg-shape similarity.

I am very glad to see a new study on this important topic. I find the results interesting but I am unclear about data processing and analyses and raised few points, which should be carefully addressed.

Page 5, line 20: Next to Luro and Hauber (2017), I would add study of Su et al (2018 Avian Research) - nest sanitation in *Pycnonotus xanthorrhous*.

P. 7, first sentence: Dimensions measured by software or manually using a calliper?

P. 7, l. 15: Add a reference of Croston and Hauber (2015 PLoS ONE) to refer on other details, such as natural egg colour measurements (e.g. colour measurements taken from active or abandoned clutches).

P. 7, l. 24: Not very clear, if these sourced data are presented as “control treatment” in this manuscript? I think it is so but when checking the control treatment in Fig. 2, I see two rejections. Is then the statement “1 rejection out of n=15 trials” correct?

P. 7, l. 34: “and/or (ii) robin eggs” change to “and/or (iii) robin eggs”

P. 8, l. 3: Is the number 1-5 treatments per nest correct? Dataset suggests 1-4.

P. 8, l. 6-8: I am missing more details on sample size. Fig. 2 provides some information but figure points are sometimes difficult to read. I would explicitly indicate number of experiments performed for each model type separately (perhaps at each model in Fig. 1B or Fig. 2 is a good place).

P. 8, l. 35: Predictor “experimental order” was modelled as continuous or categorical?

P. 9, Results: After a look on dataset, I am unclear about coding the predictor order of experiment. I expect continuous ordering starting from 1 for each nest but there many cases starting from 2 or 3. In several other cases there are gaps in ordering sequence. Further, both experiments on the nest C-022 were noted as first experiment (experiment order = 1).

I also notice marginally non-significant effect of experiment order on p. 9, l. 35. This surely deserves more attention and should be briefly discussed. For example, I see 35 observations for dependent variable in each test using data from season 2020 and thus is it possible that the power of the test is too low to detect significance? I did a quick GLMM tests in R with the same model structure and using data provided by the authors. Interestingly, I found non-significant effects for both predictors, order of experiment and model metric (panel length or width). The model outputs were relatively similar to those in manuscript only after excluding nest ID as a random effect. Am I missing something? In any case, please re-check data thoroughly and re-run analyses.

Figure 1B: Please, show the relative change in relevant variable for each model. This should be accompanied with the absolute dimensions (egg length and width, panel length) for starting and ending model at each row.

The terms “Shapeways eggs”, “Shapeways ‘cow bird’ models”, “Shapeways model egg”, “Shapeways-made model cowbird eggs”, “model cow bird egg”, “control[s]” are used interchangeably and a unifying term would help here.

Is not the person named D. Hanley in acknowledgements one of the co-authors?

Review form: Reviewer 2

Is the manuscript scientifically sound in its present form?

No

Are the interpretations and conclusions justified by the results?

No

Is the language acceptable?

Yes

Do you have any ethical concerns with this paper?

No

Have you any concerns about statistical analyses in this paper?

Yes

Recommendation?

Major revision is needed (please make suggestions in comments)

Comments to the Author(s)

This manuscript presents data regarding an interesting question for the field: what is the relationship between nest sanitation and egg rejection in brood-parasite hosts? This is an important topic as it still remains unclear how egg rejection defences arise in response to invasion by a brood-parasite. The authors use model eggs that vary from 'egg-like' to 'nest detritus-like' along two perceptual axes of variation and test rejection responses of American robins to derive response curves. This study could make a nice contribution to this growing field of examining host-defences from a more perceptual-perspective; however I found several key issues that would benefit from clarification:

(1) The rationale could be presented more convincingly.

Currently the authors introduce (i) nest hygiene behaviour and then (ii) egg rejection by brood parasite hosts, and the errors and costs associated with it. They then (iii) mention very briefly that egg rejection and detritus removal are independent behavioural traits that do not covary before (iv) stating the key question is about how birds recognise foreign eggs.

I think a more intuitive angle would be to state this question a little differently: How does egg rejection evolve? Perceptual problems are likely to be a key constraint here, and this study is a great contribution to our growing knowledge in this field. However, at the moment there is not a

strong link between presenting birds with weird looking objects in the nest and the presence/absence of egg rejection behaviour in brood parasite hosts.

How about: Egg rejection varies within and among brood parasite hosts. Thought to be mediated by costs. But an outstanding question remains - how does egg rejection initially arise? Thought to be a continuation of nest hygiene behaviour which is common across birds, but previous studies show these traits do not covary. Why might that be? Then explain why we need to look at variation in egg shape, from 'normal' to 'nest detritus' continuously.

(2) The methods lack key details

(a) The rationale for the two axes of variation is very unclear and currently requires careful reading of a different paper to understand what is being done here. The authors of the current manuscript also then label their use of the traits differently, and nowhere is it explained why or what significance this has.

(b) Similarly the model types are referred to by the name of the manufacturer rather than the biological trait of interest. It would be much easier to follow if you labelled the cowbird-like eggs as such and not 'Shapeway models', for example. On page 7, line 6 we are told that they will be referred to as 'controls' but in the next paragraph they go back to being Shapeways eggs.

(c) At what breeding stage were the eggs added to the nest? Presumably during laying and/or incubation, but this is not stated. This is critical as the timing of egg-laying by brood parasites can have large effects on the likelihood of egg rejection. How consistent was this across treatments?

(d) There are also details missing with the statistical methods and presentation of results. For example, how was experimental order treated in the models? This is critical because from previous work with a range of hosts, we know that repeated presentations of stimuli can affect rejection responses. Indeed, in this study there is a marginally non-significant effect of experiment order on the width model series (although the authors rather strongly state that it did not have an effect, which is a bit strong given the p-values were 0.056 and 0.057). There are no tables of results, or degrees of freedom stated with the chi-square statistics, so it is unclear whether this variable was continuous or categorical.

(3) Further analyses would be helpful to place these results in context.

The results of this study are very interesting in that they appear to indicate that the two axes of variation produce different shaped response curves. It was disappointing, however, that these features were not explored in more depth. For example, you could actually report the estimated inflection points (\pm error) for these curves and then use a z-test or similar to assess whether they are statistically different. Alternatively, by centering and scaling the x-axes of each plot, you could combine them into one analysis and use an interaction term to determine if the response curves really are different. This would provide more robust evidence and mean that you are not left to qualitatively compare the responses in the discussion.

(4) The discussion lacks depth.

The discussion is disappointingly brief (3 paragraphs) and lacks evaluation of what the results might mean for our understanding of how egg rejection behaviour evolves, for example, or for our understanding of perceptual limits. Instead, the discussion (i) reviews the response curve results, (ii) discusses potential confounds about egg weight and volume (with no conclusion as to whether the authors think this means their results are useful or not) and then includes (iii) a paragraph telling us that increasing numbers of studies are now looking at continuous egg characteristics and the present study is one of these. Surprisingly, there is no evaluation of whether the response curves here are similar or not to the results of Yang et al. 2019 that this study replicates, and the reader is left with no clear take-home message other than that this study uses a similar methodology to others.

Other more specific comments:

Introduction

Page 4, Line 40 - 45: this is phrased a little oddly as recognition errors are also costly if they reduce fitness. Could this be phrased differently? i.e. recognition costs can arise incidentally by damaging own eggs, or because of recognition errors. It's also pretty surprising to not see some of the classic literature on egg recognition costs referred to here (e.g. Davies et al. 1996 Proc B "Recognition errors and probability of parasitism determine whether reed warblers should accept or reject mimetic cuckoo eggs"; or Stokke 2002 Behaviour "Costs associated with recognition and rejection of parasitic eggs in two European passerines").

Page 5, line 3: this should be either "their nests" or "the nest"

Methods

Page 6, line 35: I appreciate that we can look in Iqbal et al. but could you be a bit more specific here about what 'natural plastic' you used, for replication purposes? In experiments with 3D printed stimuli, we've found that this can have major effects on rejection outcomes.

Page 6, line 40 - 55: this is a bit confusing for anyone not familiar with Yang et al. 2019. or what these terms mean. Could you perhaps introduce these dimensions a little more in the introduction? What is the significance of the 'surface-edge' variable and stereoscopic structure? Why did you label them differently in this study?

Page 8, lines 3 - 6: at what stage were these nests?

Page 8, line 35: how did you enter experimental order in your models? Is this categorical or continuous? If categorical, did you enter it as an ordered factor?

Discussion

Page 10, line 30: typo "may explain also explain"

Decision letter (RSOS-201615.R0)

Dear Dr Hauber

The Editors assigned to your paper RSOS-201615 "The limits of egg recognition: Testing the acceptance thresholds of American robins in response to decreasingly egg-shaped objects in the nest" have now received comments from reviewers and would like you to revise the paper in accordance with the reviewer comments and any comments from the Editors. Please note this decision does not guarantee eventual acceptance.

We do not generally allow multiple rounds of revision so we urge you to make every effort to fully address all of the comments at this stage. If deemed necessary by the Editors, your

manuscript will be sent back to one or more of the original reviewers for assessment. If the original reviewers are not available, we may invite new reviewers.

Please submit your revised manuscript and required files (see below) no later than 21 days from today's (ie 23-Nov-2020) date. Note: the ScholarOne system will 'lock' if submission of the revision is attempted 21 or more days after the deadline. If you do not think you will be able to meet this deadline please contact the editorial office immediately.

on behalf of Dr Kimberley Mathot (Associate Editor) and Kevin Padian (Subject Editor)
openscience@royalsociety.org

Editor comments:

I agree with the AE's decision, following the recommendations of the reviewers. Should you need more time to revise, please contact our editorial office. Thanks.

Associate Editor Comments to Author (Dr Kimberley Mathot):

Associate Editor: 1

Comments to the Author:

Your manuscript has been evaluated by two referees who are both experts in the subject area. Both referees agreed that this study addresses an important and unresolved question in brood parasitism, and believe that with appropriate revisions, the study would be suitable for publication in Royal Society Open Access.

Please revise the manuscript following the comments/suggestions from the referees, which were both thoughtful and constructive. Both referees point out that aspects of the methods require greater detail, including the statistical analyses. Both referees also pointed out the lack of discussion of results whose p values were barely > 0.05 . I would recommend discussing these (and other results) with reference not only to the p-values, but also the estimated effect size and potential biological significance. Referee #2 also provides excellent suggestions for follow-up analyses, and points out that the discussion in its current form is quite superficial.

Please be sure to provide a point-by-point response to the referees comments indicating how each have been addressed in the revised version of the manuscript.

Reviewer comments to Author:

Reviewer: 1

Comments to the Author(s)

This very interesting study tackles fundamental questions about discerning between nest hygiene and foreign egg rejection behavior in birds. The authors partly replicated methodologies of Igic et al. (2015) and Yang et al. (2019) by manipulating shape of 3D printed models on continuous scale from egg-shaped to non-egg like objects. Using the original analysis approach, they were able to detect transitions in behavioral responses according to decreasing egg-shape similarity.

I am very glad to see a new study on this important topic. I find the results interesting but I am unclear about data processing and analyses and raised few points, which should be carefully addressed.

Page 5, line 20: Next to Luro and Hauber (2017), I would add study of Su et al (2018 Avian Research) - nest sanitation in *Pycnonotus xanthorrhous*.

P. 7, first sentence: Dimensions measured by software or manually using a calliper?

P. 7, l. 15: Add a reference of Croston and Hauber (2015 PLoS ONE) to refer on other details, such as natural egg colour measurements (e.g. colour measurements taken from active or abandoned clutches).

P. 7, l. 24: Not very clear, if these sourced data are presented as "control treatment" in this manuscript? I think it is so but when checking the control treatment in Fig. 2, I see two rejections. Is then the statement "1 rejection out of n=15 trials" correct?

P. 7, l. 34: "and/or (ii) robin eggs" change to "and/or (iii) robin eggs"

P. 8, l. 3: Is the number 1-5 treatments per nest correct? Dataset suggests 1-4.

P. 8, l. 6-8: I am missing more details on sample size. Fig. 2 provides some information but figure points are sometimes difficult to read. I would explicitly indicate number of experiments performed for each model type separately (perhaps at each model in Fig. 1B or Fig. 2 is a good place).

P. 8, l. 35: Predictor "experimental order" was modelled as continuous or categorical?

P. 9, Results: After a look on dataset, I am unclear about coding the predictor order of experiment. I expect continuous ordering starting from 1 for each nest but there many cases starting from 2 or 3. In several other cases there are gaps in ordering sequence. Further, both experiments on the nest C-022 were noted as first experiment (experiment order = 1).

I also notice marginally non-significant effect of experiment order on p. 9, l. 35. This surely deserves more attention and should be briefly discussed. For example, I see 35 observations for dependent variable in each test using data from season 2020 and thus is it possible that the power of the test is too low to detect significance? I did a quick GLMM tests in R with the same model structure and using data provided by the authors. Interestingly, I found non-significant effects for both predictors, order of experiment and model metric (panel length or width). The model outputs were relatively similar to those in manuscript only after excluding nest ID as a random effect. Am I missing something? In any case, please re-check data thoroughly and re-run analyses.

Figure 1B: Please, show the relative change in relevant variable for each model. This should be accompanied with the absolute dimensions (egg length and width, panel length) for starting and ending model at each row.

The terms “Shapeways eggs”, “Shapeways ‘cow bird’ models”, “Shapeways model egg”, “Shapeways-made model cowbird eggs”, “model cow bird egg”, “control[s]” are used interchangeably and a unifying term would help here.

Is not the person named D. Hanley in acknowledgements one of the co-authors?

Reviewer: 2

Comments to the Author(s)

This manuscript presents data regarding an interesting question for the field: what is the relationship between nest sanitation and egg rejection in brood-parasite hosts? This is an important topic as it still remains unclear how egg rejection defences arise in response to invasion by a brood-parasite. The authors use model eggs that vary from 'egg-like' to 'nest detritus-like' along two perceptual axes of variation and test rejection responses of American robins to derive response curves. This study could make a nice contribution to this growing field of examining host-defences from a more perceptual-perspective; however I found several key issues that would benefit from clarification:

(1) The rationale could be presented more convincingly.

Currently the authors introduce (i) nest hygiene behaviour and then (ii) egg rejection by brood parasite hosts, and the errors and costs associated with it. They then (iii) mention very briefly that egg rejection and detritus removal are independent behavioural traits that do not covary before (iv) stating the key question is about how birds recognise foreign eggs.

I think a more intuitive angle would be to state this question a little differently: How does egg rejection evolve? Perceptual problems are likely to be a key constraint here, and this study is a great contribution to our growing knowledge in this field. However, at the moment there is not a strong link between presenting birds with weird looking objects in the nest and the presence/absence of egg rejection behaviour in brood parasite hosts.

How about: Egg rejection varies within and among brood parasite hosts. Thought to be mediated by costs. But an outstanding question remains - how does egg rejection initially arise? Thought to be a continuation of nest hygiene behaviour which is common across birds, but previous studies show these traits do not covary. Why might that be? Then explain why we need to look at variation in egg shape, from 'normal' to 'nest detritus' continuously.

(2) The methods lack key details

(a) The rationale for the two axes of variation is very unclear and currently requires careful reading of a different paper to understand what is being done here. The authors of the current manuscript also then label their use of the traits differently, and nowhere is it explained why or what significance this has.

(b) Similarly the model types are referred to by the name of the manufacturer rather than the biological trait of interest. It would be much easier to follow if you labelled the cowbird-like eggs as such and not 'Shapeway models', for example. On page 7, line 6 we are told that they will be referred to as 'controls' but in the next paragraph they go back to being Shapeways eggs.

(c) At what breeding stage were the eggs added to the nest? Presumably during laying and/or incubation, but this is not stated. This is critical as the timing of egg-laying by brood parasites can have large effects on the likelihood of egg rejection. How consistent was this across treatments?

(d) There are also details missing with the statistical methods and presentation of results. For example, how was experimental order treated in the models? This is critical because from previous work with a range of hosts, we know that repeated presentations of stimuli can affect rejection responses. Indeed, in this study there is a marginally non-significant effect of experiment order on the width model series (although the authors rather strongly state that it did

not have an effect, which is a bit strong given the p-values were 0.056 and 0.057). There are no tables of results, or degrees of freedom stated with the chi-square statistics, so it is unclear whether this variable was continuous or categorical.

(3) Further analyses would be helpful to place these results in context.

The results of this study are very interesting in that they appear to indicate that the two axes of variation produce different shaped response curves. It was disappointing, however, that these features were not explored in more depth. For example, you could actually report the estimated inflection points (\pm error) for these curves and then use a z-test or similar to assess whether they are statistically different. Alternatively, by centering and scaling the x-axes of each plot, you could combine them into one analysis and use an interaction term to determine if the response curves really are different. This would provide more robust evidence and mean that you are not left to qualitatively compare the responses in the discussion.

(4) The discussion lacks depth.

The discussion is disappointingly brief (3 paragraphs) and lacks evaluation of what the results might mean for our understanding of how egg rejection behaviour evolves, for example, or for our understanding of perceptual limits. Instead, the discussion (i) reviews the response curve results, (ii) discusses potential confounds about egg weight and volume (with no conclusion as to whether the authors think this means their results are useful or not) and then includes (iii) a paragraph telling us that increasing numbers of studies are now looking at continuous egg characteristics and the present study is one of these. Surprisingly, there is no evaluation of whether the response curves here are similar or not to the results of Yang et al. 2019 that this study replicates, and the reader is left with no clear take-home message other than that this study uses a similar methodology to others.

Other more specific comments:

Introduction

Page 4, Line 40 - 45: this is phrased a little oddly as recognition errors are also costly if they reduce fitness. Could this be phrased differently? i.e. recognition costs can arise incidentally by damaging own eggs, or because of recognition errors. It's also pretty surprising to not see some of the classic literature on egg recognition costs referred to here (e.g. Davies et al. 1996 Proc B "Recognition errors and probability of parasitism determine whether reed warblers should accept or reject mimetic cuckoo eggs"; or Stokke 2002 Behaviour "Costs associated with recognition and rejection of parasitic eggs in two European passerines").

Page 5, line 3: this should be either "their nests" or "the nest"

Methods

Page 6, line 35: I appreciate that we can look in Iqbal et al. but could you be a bit more specific here about what 'natural plastic' you used, for replication purposes? In experiments with 3D printed stimuli, we've found that this can have major effects on rejection outcomes.

Page 6, line 40 - 55: this is a bit confusing for anyone not familiar with Yang et al. 2019. or what these terms mean. Could you perhaps introduce these dimensions a little more in the introduction? What is the significance of the 'surface-edge' variable and stereoscopic structure? Why did you label them differently in this study?

Page 8, lines 3 - 6: at what stage were these nests?

Page 8, line 35: how did you enter experimental order in your models? Is this categorical or continuous? If categorical, did you enter it as an ordered factor?

Discussion

Page 10, line 30: typo “may explain also explain”

===PREPARING YOUR MANUSCRIPT===

===PREPARING YOUR REVISION IN SCHOLARONE===

- 1) One version identifying all the changes that have been made (for instance, in coloured highlight, in bold text, or tracked changes);
 - 2) A 'clean' version of the new manuscript that incorporates the changes made, but does not highlight them.
 - An individual file of each figure (EPS or print-quality PDF preferred [either format should be produced directly from original creation package], or original software format).
 - An editable file of each table (.doc, .docx, .xls, .xlsx, or .csv).
 - An editable file of all figure and table captions.
- Note: you may upload the figure, table, and caption files in a single Zip folder.
- Any electronic supplementary material (ESM).
 - If you are requesting a discretionary waiver for the article processing charge, the waiver form must be included at this step.
 - If you are providing image files for potential cover images, please upload these at this step, and inform the editorial office you have done so. You must hold the copyright to any image provided.
 - A copy of your point-by-point response to referees and Editors. This will expedite the preparation of your proof.

- Ensure that your data access statement meets the requirements at <https://royalsociety.org/journals/authors/author-guidelines/#data>. You should ensure that you cite the dataset in your reference list. If you have deposited data etc in the Dryad repository, please include both the 'For publication' link and 'For review' link at this stage.
- If you are requesting an article processing charge waiver, you must select the relevant waiver option (if requesting a discretionary waiver, the form should have been uploaded at Step 3 'File upload' above).
- If you have uploaded ESM files, please ensure you follow the guidance at <https://royalsociety.org/journals/authors/author-guidelines/#supplementary-material> to include a suitable title and informative caption. An example of appropriate titling and captioning may be found at [https://figshare.com/articles/Table_S2_from_Is_there_a_trade-off_between_peak_performance_and_performance_breadth_across_temperatures_for_aerobic_sc ope_in_teleost_fishes_/3843624](https://figshare.com/articles/Table_S2_from_Is_there_a_trade-off_between_peak_performance_and_performance_breadth_across_temperatures_for_aerobic_scope_in_teleost_fishes_/3843624).

Author's Response to Decision Letter for (RSOS-201615.R0)

See Appendix A.

Decision letter (RSOS-201615.R1)

Dear Dr Hauber

On behalf of the Editors, we are pleased to inform you that your Manuscript RSOS-201615.R1 "The limits of egg recognition: Testing acceptance thresholds of American robins in response to decreasingly egg-shaped objects in the nest" has been accepted for publication in Royal Society Open Science subject to minor revision in accordance with the referees' reports. Please find the feedback from the Editors below my signature.

We invite you to respond to the comments and revise your manuscript. Below the Editors' comments (where applicable) we provide additional requirements. Final acceptance of your manuscript is dependent on these requirements being met. We provide guidance below to help you prepare your revision.

Please submit your revised manuscript and required files (see below) no later than 7 days from today's (ie 11-Dec-2020) date. Note: the ScholarOne system will 'lock' if submission of the revision is attempted 7 or more days after the deadline. If you do not think you will be able to meet this deadline please contact the editorial office immediately.

on behalf of Dr Kimberley Mathot (Associate Editor) and Kevin Padian (Subject Editor)
openscience@royalsociety.org

Associate Editor Comments to Author (Dr Kimberley Mathot):

Thank you for submitting your revised manuscript. I feel that you have done a thorough job addressing the comments received from the referees in the initial review of the manuscript. I have just one additional revision I would like to request, which has to do with the use of the term "treatment order". When you have a series of up to 4 treatments per nest (A, B, C, and D), treatment order implies a categorical variable such as ABCD, ABDC, ACDB, etc.). Can you instead refer to "treatment number" throughout the manuscript, and explicitly state in the methods that treatment number refers to the sequence with which the treatment occurred at a given nest.

===PREPARING YOUR MANUSCRIPT===

===PREPARING YOUR REVISION IN SCHOLARONE===

- Any electronic supplementary material (ESM).
- If you are requesting a discretionary waiver for the article processing charge, the waiver form must be included at this step.
- If you are providing image files for potential cover images, please upload these at this step, and inform the editorial office you have done so. You must hold the copyright to any image provided.
- A copy of your point-by-point response to referees and Editors. This will expedite the preparation of your proof.

- Ensure that your data access statement meets the requirements at <https://royalsociety.org/journals/authors/author-guidelines/#data>. You should ensure that you cite the dataset in your reference list. If you have deposited data etc in the Dryad repository, please only include the 'For publication' link at this stage. You should remove the 'For review' link.
- If you are requesting an article processing charge waiver, you must select the relevant waiver option (if requesting a discretionary waiver, the form should have been uploaded at Step 3 'File upload' above).
- If you have uploaded ESM files, please ensure you follow the guidance at <https://royalsociety.org/journals/authors/author-guidelines/#supplementary-material> to include a suitable title and informative caption. An example of appropriate titling and captioning may be found at https://figshare.com/articles/Table_S2_from_Is_there_a_trade-off_between_peak_performance_and_performance_breadth_across_temperatures_for_aerobic_scope_in_teleost_fishes_/3843624.

Author's Response to Decision Letter for (RSOS-201615.R1)

See Appendix B.

Decision letter (RSOS-201615.R2)

This year has been very difficult for everyone, and we want to take the opportunity to thank you for your continued support in 2020.

The Royal Society Open Science editorial office will be closed from the evening of Friday 18 December 2020 until Monday 4 January 2021. We will not be responding during this time. If you have received a deadline within this time period, please contact us as soon as possible to allow us to extend the deadline. If you receive any automated messages during this time asking you to meet a deadline, we offer apologies and invite you to respond after the festive period or during normal working hours.

With our best for a peaceful festive period and New Year, and we look forward to working with you in 2021.

Dear Dr Hauber,

It is a pleasure to accept your manuscript entitled "The limits of egg recognition: Testing acceptance thresholds of American robins in response to decreasingly egg-shaped objects in the nest" in its current form for publication in Royal Society Open Science.

on behalf of Dr Kimberley Mathot (Associate Editor) and Kevin Padian (Subject Editor)
openscience@royalsociety.org

Appendix A

Dear Profs. Mathot and Padian:

Thank you for your continued interest in our manuscript and for the referees' constructive comments on its content.

Below we explain how we revised our text according to the criticism and we trust that the new draft will be suitable for consideration to publish in Royal Society Open Science.

Thank you, sincerely, Mark Hauber and coauthors.

Dear Dr Hauber

The Editors assigned to your paper RSOS-201615 "The limits of egg recognition: Testing the acceptance thresholds of American robins in response to decreasingly egg-shaped objects in the nest" have now received comments from reviewers and would like you to revise the paper in accordance with the reviewer comments and any comments from the Editors. Please note this decision does not guarantee eventual acceptance.

OUR RESPONSE: We have revised the ms in detail in light of the comments and explain how we did so point-by-point.

[...]

on behalf of Dr Kimberley Mathot (Associate Editor) and Kevin Padian (Subject Editor)
openscience@royalsociety.org

Editor comments:

I agree with the AE's decision, following the recommendations of the reviewers. Should you need more time to revise, please contact our editorial office. Thanks.

Associate Editor Comments to Author (Dr Kimberley Mathot):

Associate Editor: 1

Comments to the Author:

Your manuscript has been evaluated by two referees who are both experts in the subject area. Both referees agreed that this study addresses an important and unresolved question in brood parasitism, and believe that with appropriate revisions, the study would be suitable for publication in Royal Society Open Access.

Our Response: Thank you for your positive assessment.

Please revise the manuscript following the comments/suggestions from the referees, which were both thoughtful and constructive. Both referees point out that aspects of the methods require greater detail, including the statistical analyses.

Our Response: Please note that we have detected a fatal flaw in JMP 12.0 Software during our revision. Namely, we discovered that JMP 12.0 appears to conduct random-effects binomial tests but in fact, it simply drops the random effect from the model. Therefore, we re-did all of our statistics in the R statistical environment. We lost one significant result but the rest of the results and patterns remained statistically the same.

Both referees also pointed out the lack of discussion of results whose p values were barely > 0.05. I would recommend discussing these (and other results) with reference not only to the p-values, but also the estimated effect size and potential biological significance.

Our Response: We now fully discuss potential biological significance of both marginally significant and non-significant results. We also present odds ratios for each response variable for each dataset to allow for an assessment of relative effect sizes and their biological impact.

Referee #2 also provides excellent suggestions for follow-up analyses, and points out that the discussion in its current form is quite superficial.

Our Response: To address these concerns, the discussion is now expanded; also please see below.

Please be sure to provide a point-by-point response to the referees comments indicating how each have been addressed in the revised version of the manuscript.

Our Response: We now explain our changes and responses as per below.

Reviewer comments to Author:

Reviewer: 1

Comments to the Author(s)

This very interesting study tackles fundamental questions about discerning between nest hygiene and foreign egg rejection behavior in birds. The authors partly replicated methodologies of Igic et al. (2015) and Yang et al. (2019) by manipulating shape of 3D printed models on continuous scale from egg-shaped to non-egg like objects. Using the original analysis approach, they were able to detect transitions in behavioral responses according to decreasing egg-shape similarity.

I am very glad to see a new study on this important topic. I find the results interesting but I am unclear about data processing and analyses and raised few points, which should be carefully addressed.

Our Response: Thank you very much for your interests and for the opportunity to clarify our methods and results.

Page 5, line 20: Next to Luro and Hauber (2017), I would add study of Su et al (2018 Avian Research) - nest sanitation in *Pycnonotus xanthorrhous*.

Our Response: Thank you, replaced.

P. 7, first sentence: Dimensions measured by software or manually using a calliper?

Our Response: Explained now that it was done by a caliper to the nearest 0.1 mm.

P. 7, l. 15: Add a reference of Croston and Hauber (2015 PLoS ONE) to refer on other details, such as natural egg colour measurements (e.g. colour measurements taken from active or abandoned clutches).

Our Response: We added Hauber et al. 2020b which is based on our most recent measurements of natural robin eggs.

P. 7, l. 24: Not very clear, if these sourced data are presented as "control treatment" in this manuscript? I think it is so but when checking the control treatment in Fig. 2, I see two rejections. Is then the statement "1 rejection out of n=15 trials" correct?

Our Response: Corrected in the figure.

P. 7, l. 34: "and/or (ii) robin eggs" change to "and/or (iii) robin eggs"

Our Response: Done.

P. 8, l. 3: Is the number 1–5 treatments per nest correct? Dataset suggests 1–4.

Our Response: Corrected.

P. 8, l. 6–8: I am missing more details on sample size. Fig. 2 provides some information but figure points are sometimes difficult to read. I would explicitly indicate number of experiments performed for each model type separately (perhaps at each model in Fig. 1B or Fig. 2 is a good place).

Our Response: Done, sample sizes per treatments are specified in the Fig. 2AB legend.

P. 8, l. 35: Predictor "experimental order" was modelled as continuous or categorical?

Our Response: Now stated as continuous variable.

P. 9, Results: After a look on dataset, I am unclear about coding the predictor order of experiment. I expect continuous ordering starting from 1 for each nest but there many cases starting from 2 or 3. In several other cases there are gaps in ordering sequence.

Our Response: Please note that nests with predation ($n=6$) or abandonment ($n=8$) and, thus, no accept/reject outcome were also present in the original dataset's ordering, but these are not included in the final data set as nest predation is stochastic whereas nest abandonment is not a response by American Robins to experimental brood parasitism (Croston and Hauber 2014).

Further, both experiments on the nest C-022 were noted as first experiment (experiment order = 1).

Our Response: This is corrected now.

I also notice marginally non-significant effect of experiment order on p. 9, l. 35. This surely deserves more attention and should be briefly discussed. For example, I see 35 observations for dependent variable in each test using data from season 2020 and thus is it possible that the power of the test is too low to detect significance? I did a quick GLMM tests in R with the same model structure and using data provided by the authors. Interestingly, I found non-significant effects for both predictors, order of experiment and model metric (panel

length or width). The model outputs were relatively similar to those in manuscript only after excluding nest ID as a random effect. Am I missing something? In any case, please re-check data thoroughly and re-run analyses.

Our Response: Thank you. As stated above, please note that we have detected a fatal flaw in JMP 12.0 Software during our revision. Namely, we discovered that JMP 12.0 appears to conduct random-effects binomial tests but in fact, it simply drops the random effect from the model output (which is why you also found what we found when you excluded the random effects). Therefore, we re-did all of our statistics in the R statistical environment. We only lost one significant result but the rest of the results and patterns remained statistically the same. Nonetheless, we now also discuss marginally non-significant findings and their potential biological significance in light of effect sizes.

Figure 1B: Please, show the relative change in relevant variable for each model. This should be accompanied with the absolute dimensions (egg length and width, panel length) for starting and ending model at each row.

Our Response: We include the starting length and widths for our model eggs in the Methods and for the models in the Figure Legend of Fig. 1B.

The terms "Shapeways eggs", "Shapeways 'cow bird' models", "Shapeways model egg", "Shapeways-made model cowbird eggs", "model cow bird egg", "control[s]" are used interchangeably and a unifying term would help here.

Our Response: After the introductory description and manufacture, we call these eggs "control eggs" thereafter.

Is not the person named D. Hanley in acknowledgements one of the co-authors?

Our Response: Correct, now deleted.

Reviewer: 2

Comments to the Author(s)

This manuscript presents data regarding an interesting question for the field: what is the relationship between nest sanitation and egg rejection in brood-parasite hosts? This is an important topic as it still remains unclear how egg rejection defences arise in response to invasion by a brood-parasite. The authors use model eggs that vary from 'egg-like' to 'nest detritus-like' along two perceptual axes of variation and test rejection responses of American robins to derive response curves. This study could make a nice contribution to this growing field of examining host-defences from a more perceptual-perspective; however I found several key issues that would benefit from clarification:

Our Response: Thank you for your detailed interest in our study.

(1) The rationale could be presented more convincingly.

Currently the authors introduce (i) nest hygiene behaviour and then (ii) egg rejection by brood parasite hosts, and the errors and costs associated with it. They then (iii) mention very briefly that egg rejection and detritus removal are independent behavioural traits that do not covary before (iv) stating the key question is about how birds recognise foreign eggs.

I think a more intuitive angle would be to state this question a little differently: How does egg rejection evolve? Perceptual problems are likely to be a key constraint here, and this study is a great contribution to our growing knowledge in this field.

Our Response: Agreed, we are very interested in the evolutionary origins of egg rejection per se. We now state so in both the Abstract and keep this statement in the Introduction too.

However, at the moment there is not a strong link between presenting birds with weird looking objects in the nest and the presence/absence of egg rejection behaviour in brood parasite hosts.

How about: Egg rejection varies within and among brood parasite hosts. Thought to be mediated by costs. But an outstanding question remains - how does egg rejection initially arise? Thought to be a continuation of nest hygiene behaviour which is common across birds, but previous studies show these traits do not covary. Why might that be? Then explain why we need to look at variation in egg shape, from 'normal' to 'nest detritus' continuously.

Our Response: Thank you, we have now revised our Introduction along these lines.

(2) The methods lack key details

(a) The rationale for the two axes of variation is very unclear and currently requires careful reading of a different paper to understand what is being done here. The authors of the current manuscript also then label their use of the traits differently, and nowhere is it explained why or what significance this has.

Our Response: We now introduce our two axes of variation strongly and provide our definitions and consistent terminology.

(b) Similarly the model types are referred to by the name of the manufacturer rather than the biological trait of interest. It would be much easier to follow if you labelled the cowbird-like eggs as such and not 'Shapeway models', for example. On page 7, line 6 we are told

that they will be referred to as 'controls' but in the next paragraph they go back to being Shapeways eggs.

Our Response: After the introductory description and manufacture, we call these eggs "control eggs" thereafter.

(c) At what breeding stage were the eggs added to the nest? Presumably during laying and/or incubation, but this is not stated. This is critical as the timing of egg-laying by brood parasites can have large effects on the likelihood of egg rejection. How consistent was this across treatments?

Our Response: We now explain: *These methods yielded nests for experimental treatments during the laying and incubation stages, for which we now specify that the data were combined based on a lack of difference in egg rejection rates between these stages during our own prior experiments on robins at this study site (Abolins-Abols and Hauber 2020).*

(d) There are also details missing with the statistical methods and presentation of results. For example, how was experimental order treated in the models? This is critical because from previous work with a range of hosts, we know that repeated presentations of stimuli can affect rejection responses.

Our Response: In several of our previous works on robins we did not find an order effect when experimenting repeatedly on the same individual robin (e.g. Croston and Hauber 2014, Aidala et al. 2015); nonetheless, we included experiment order here as a continuous variable. This was important here since this time we did detect (and now discuss) significant patterns. We thank you.

Indeed, in this study there is a marginally non-significant effect of experiment order on the width model series (although the authors rather strongly state that it did not have an effect, which is a bit strong given the p-values were 0.056 and 0.057). There are no tables of results, or degrees of freedom stated with the chi-square statistics, so it is unclear whether this variable was continuous or categorical.

Our Response: We have completely redone our statistics because of a fatal error that we found in the JMP 12.0 statistical software regarding the analysis of random effects binomial models. We now redid all the analyses in the R statistical environment, therefore; the main statistical results remained consistent, but we now discuss marginally significant/non-significant effects, including the impact of order, too, in our expanded Discussion

(3) Further analyses would be helpful to place these results in context. The results of this study are very interesting in that they appear to indicate that the two axes of variation produce different shaped response curves. It was disappointing, however, that

these features were not explored in more depth. For example, you could actually report the estimated inflection points (\pm error) for these curves and then use a z-test or similar to assess whether they are statistically different. Alternatively, by centering and scaling the x-axes of each plot, you could combine them into one analysis and use an interaction term to determine if the response curves really are different. This would provide more robust evidence and mean that you are not left to qualitatively compare the responses in the discussion.

Our Response: We now discuss these issues more extensively in the Discussion in a new section by looking at a supplementary figure, where the raw-values of the x-axes are replaced by z-score transformed metrics.

(4) The discussion lacks depth.

The discussion is disappointingly brief (3 paragraphs) and lacks evaluation of what the results might mean for our understanding of how egg rejection behaviour evolves, for example, or for our understanding of perceptual limits.

Our Response: We have expanded the Discussion by substantive and concluding paragraphs now.

Instead, the discussion (i) reviews the response curve results, (ii) discusses potential confounds about egg weight and volume (with no conclusion as to whether the authors think this means their results are useful or not) and then includes (iii) a paragraph telling us that increasing numbers of studies are now looking at continuous egg characteristics and the present study is one of these. Surprisingly, there is no evaluation of whether the response curves here are similar or not to the results of Yang et al. 2019 that this study replicates,

Our Response: We now compare our results to those of Yang et al. 2019 in a full paragraph.

and the reader is left with no clear take-home message other than that this study uses a similar methodology to others.

Our Response: We have expanded upon our concluding paragraph now.

Other more specific comments:

Introduction

Page 4, Line 40 - 45: this is phrased a little oddly as recognition errors are also costly if they reduce fitness. Could this be phrased differently? i.e. recognition costs can arise incidentally by damaging own eggs, or because of recognition errors. It's also pretty surprising to not see some of the classic literature on egg recognition costs referred to here (e.g. Davies et al.

1996 Proc B "Recognition errors and probability of parasitism determine whether reed warblers should accept or reject mimetic cuckoo eggs"; or Stokke 2002 Behaviour "Costs associated with recognition and rejection of parasitic eggs in two European passerines").

Our Response: We now provide a clearer explanation, including the cost of accidentally damaging the host's own eggs; and we also cite these 2 classic papers.

Page 5, line 3: this should be either "their nests" or "the nest"

Our Response: Corrected.

Methods

Page 6, line 35: I appreciate that we can look in Igic et al. but could you be a bit more specific here about what 'natural plastic' you used, for replication purposes? In experiments with 3D printed stimuli, we've found that this can have major effects on rejection outcomes.

Our Response: The plastic material is explained for both types of eggs sourced. Also, we state that we painted our eggs with 3 coats of paint which we now explain that it removed any surface-effect of the plastic and standardize it by paint coat consistency.

Page 6, line 40 - 55: this is a bit confusing for anyone not familiar with Yang et al. 2019. or what these terms mean. Could you perhaps introduce these dimensions a little more in the introduction? What is the significance of the 'surface-edge' variable and stereoscopic structure? Why did you label them differently in this study?

Our Response: More explanation and definitions are provided. The terminology is also made simpler and more consistent throughout the text.

Page 8, lines 3 - 6: at what stage were these nests?

Our Response: Clarified two paragraphs above now (as above, too).

Page 8, line 35: how did you enter experimental order in your models? Is this categorical or continuous? If categorical, did you enter it as an ordered factor?

Our Response: Continuous variable; now explicitly stated.

Discussion

Page 10, line 30: typo "may explain also explain"

Our Response: Corrected.

===PREPARING YOUR MANUSCRIPT===

- one version identifying all the changes that have been made (for instance, in coloured highlight, in bold text, or tracked changes);
- a 'clean' version of the new manuscript that incorporates the changes made, but does not highlight them. This version will be used for typesetting if your manuscript is accepted.

===PREPARING YOUR REVISION IN SCHOLARONE===

Attach your point-by-point response to referees and Editors at Step 1 'View and respond to

decision letter'. This document should be uploaded in an editable file type (.doc or .docx are preferred). This is essential.

-- If you have uploaded ESM files, please ensure you follow the guidance at <https://royalsociety.org/journals/authors/author-guidelines/#supplementary-material> to include a suitable title and informative caption. An example of appropriate titling and

captioning may be found at https://figshare.com/articles/Table_S2_from_Is_there_a_trade-off_between_peak_performance_and_performance_breadth_across_temperatures_for_aerobic_scope_in_teleost_fishes_/3843624.

Appendix B

Dear RSOS Editorial Board Members, dear Prof. Mathot,

Thank you for the positive evaluation of our revised ms. We have attached below a marked up document where we:

1. Removed all reference to “treatment order”, and replaced it with “trial number”.
2. We also explain this variable’s definition in the Methods as: *The response variable was the bivariate outcome of each trial (accepted/rejected), with model-series trait metric (panel length or width) (each as a continuous trait) and trial number (whether 1st, 2nd, etc. per each nest) of the experiment (also as a continuous trait) included as predictor variables and nest ID used as a random effect.*
3. We thank the editors and the referees of the journal for their comments in the Acknowledgement now.

Thank you, sincerely, Mark Hauber and coauthors.